# Effect of Morphological Modification over g-C$_3$N$_4$ on Photocatalytic Hydrogen Evolution Performance of g-C$_3$N$_4$-Pt Photocatalysts

Thi Van Anh Hoang [ID], Phuong Anh Nguyen and Eun Woo Shin *[ID]

School of Chemical Engineering, University of Ulsan, Ulsan 44610, Republic of Korea
* Correspondence: ewshin@ulsan.ac.kr; Tel.: +82-52-259-2253

**Abstract:** In this study, the morphological properties of g-C$_3$N$_4$ in g-C$_3$N$_4$-Pt photocatalysts were modified by a simple hydrothermal treatment for photocatalytic hydrogen evolution. In addition, the morphological modification effect of g-C$_3$N$_4$ on the hydrogen evolution performance was investigated. The long-time hydrothermal treatment clearly changed the morphology of g-C$_3$N$_4$ by building extended melem units with more oxygen functional groups at the defect edges of the extended melem units, which was evidenced by X-ray diffraction (XRD), Fourier transform infrared spectroscopy (FTIR), and X-ray photoelectron spectroscopy (XPS) measurements. The different morphological features of g-C$_3$N$_4$ resulted in lower photoluminescence (PL) emission intensity in PL spectra and a smaller semicircle radius in electrochemical impedance spectroscopy (EIS) data. This indicates the more efficient charge separation of the g-C$_3$N$_4$-Pt photocatalyst with a modified morphology. Consequently, morphologically modified g-C$_3$N$_4$-Pt showed a higher photocatalytic hydrogen evolution rate due to the better charge separation efficiency.

**Keywords:** photocatalyst; g-C$_3$N$_4$-Pt; hexagonal rod-like structure; tri-s-triazine unit; hydrogen evolution





## 1. Introduction

A sunlight-driven hydrogen evolution process from water is considered a long-term sustainable hydrogen production technology with no CO$_2$ emissions. This is because sunlight and water are clean, abundant, and renewable sources [1–3]. Various semiconductor photocatalysts have been widely tested to overcome the limited photocatalytic hydrogen evolution efficiency, which is a challenging task [4–7]. Among various semiconductor photocatalysts, graphitic carbon nitride (g-C$_3$N$_4$) has attracted great interest since metal-free polymeric semiconductors have several advantages, such as low-cost precursors and ecological friendliness, high physicochemical stability, and a sufficient band gap for visible-light utilization [8,9]. Although enormous efforts have been made in various ways to utilize photocatalysts, there are still many challenging problems in the photocatalytic hydrogen evolution of g-C$_3$N$_4$ [10–12].

In recent years, the defect engineering and morphological modification of g-C$_3$N$_4$ have been extensively employed to significantly enhance its photocatalytic performance [13–16]. A simple solvothermal process has been introduced as an effective treatment to modify its morphological properties, including by creating defect sites in g-C$_3$N$_4$, resulting in high photocatalytic activity [17–20]. However, even though the solvothermal treatment is a facile and green method for generating desirable defect sites and oxygen functional groups on the g-C$_3$N$_4$ surface, the morphological modification of the g-C$_3$N$_4$ structure, including the extension of melem units, has also been observed under harsh treatment conditions [21,22]. To the best of our knowledge, there have been no reports on the effects of extended melem unit formation on g-C$_3$N$_4$ via hydrothermal treatment, Pt properties, or the hydrogen evolution performance of g-C$_3$N$_4$-Pt photocatalysts.

In this study, we investigated the morphological effect of the extended melem units formed over g-C$_3$N$_4$ on the photocatalytic performance of g-C$_3$N$_4$-Pt photocatalysts. The morphologically modified g-C$_3$N$_4$-Pt photocatalyst was prepared by a simple hydrothermal treatment, and photocatalytic hydrogen evolution tests were conducted to compare the hydrogen evolution activity of the prepared photocatalysts. In addition, the properties of the prepared photocatalysts were characterized by various techniques. The long-term hydrothermal treatment induced the morphological modification of g-C$_3$N$_4$ by forming extended melem units and introducing oxygen functional groups into the defect sites of the extended melem units. Photoluminescence (PL) spectra, electrochemical impedance spectroscopy (EIS) data, and the photocatalytic hydrogen evolution results clearly demonstrated that morphologically modified g-C$_3$N$_4$-Pt had higher charge separation efficiency, resulting in better photocatalytic hydrogen evolution performance.

## 2. Results and Discussion

### 2.1. Physicochemical Properties

The X-ray diffraction (XRD) patterns of the photocatalysts are shown in Figure 1A. For CN0-Pt and CN3-Pt, there were two typical diffraction peaks of the CN structure located at 2θ values of 13.0 and 27.7°. The weak peak at the 2θ value of 13.0° is ascribed to recurring (100) in-plane structural motifs in g-C$_3$N$_4$, and the strong diffraction peak at the 2θ value of 27.7° is related to (002) interplanar stacking reflection [23–25]. For CN9-Pt, the characteristic peak of CN at 2θ of 27.7° is assigned to the (002) crystal plane. Additionally, the shift from the slight peak at the 2θ of 13.0° to the intense peak at 10.6° demonstrates the extension of the tri-s-triazine unit (melem unit) in the CN structure after long-time hydrothermal treatment, effectively influencing the morphological structure of CN9-Pt, and will be clarified by field-emission scanning electron microscopy (FE-SEM) images [21,26]. The higher intensity of the peaks at 2θ values of 10.6 and 27.7° for CN9-Pt compared to CN0-Pt and CN3-Pt indicates improved crystallinity and an increased charge-transfer efficiency, consequently enhancing the photocatalytic activity [27]. The peaks at 2θ values of 39.7 and 46.4° correspond to the (111) and (200) crystal planes of Pt (JCPDS: 87-0636), revealing successful Pt loading on g-C$_3$N$_4$ materials [28,29]. The Fourier transform infrared spectroscopy (FTIR) spectra of CN0-Pt, CN3-Pt, and CN9-Pt are shown in Figure 1B. The broad peak in the range of 3000–3500 cm$^{-1}$ is due to the single-bond interaction with hydrogen (hydroxyl groups or NHx groups) [30]. The typical stretching mode of aromatic C = N or C-N was located at around 1200–1700 cm$^{-1}$ [31,32]. The peak at 810 cm$^{-1}$ refers to the heptazine cycle present in the CN structure [33]. It was confirmed that the main chemical structure of CN was relatively maintained after the long-time hydrothermal process.

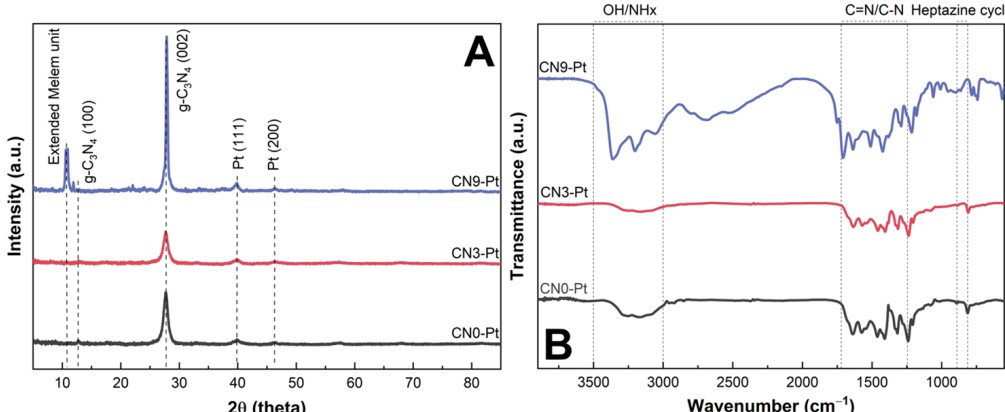

**Figure 1.** (**A**) XRD patterns and (**B**) FTIR spectra of CNx-Pt samples.

X-ray photoelectron spectroscopy (XPS) was studied and is shown in Figure 2 to further confirm the composition and surface chemical status of CNx-Pt after a long-time hydrothermal process. The C 1s spectra of the as-prepared samples could be deconvoluted into four peaks with binding energies of 284.5, 286.3, 287.7, and 289.0 eV, ascribed to C–C species, C–NHx, N = C–N, and COOH coordination of CN, respectively [34–36]. Furthermore, there was no apparent shift in the C 1s peak in CNx-Pt, indicating that the $sp^2$-bonded carbon in CNx-Pt was stable, as determined by FTIR. The N 1s spectra comprised three peaks at 398.1, 399.5, and 400.6 eV, corresponding to N bonded in the heptazine ring of CN and the existence of tertiary N–$(C)_3$ groups and amino–NHx groups [37,38]. The deconvoluted peak of O 1s spectra could be well fitted to peaks at 530.6, 531.0, 532.2, and 533.0 eV. The peak at 531.0 eV can be assigned to the PtO group [39], and the three other peaks are ascribed to the COOH, C = O, and OH groups, respectively [40,41]. As shown in Table 1, the atomic percentage of C = O species in CNx-Pt was dramatically reduced, whereas the COOH group revealed the growth of the atomic percentage. The increase in COOH was 13.86% (CN0-Pt) < 25.85 at% (CN3-Pt) < 63.46 at% (CN9-Pt), and the decrease in C = O was 71.57 at% (CN0-Pt) > 39.89 at% (CN3-Pt) > 25.13 at% (CN9-Pt). As a result, it seems that the existence of COOH in CN could be due to the conversion of C = O species after the long-time hydrothermal treatment. In addition, a significantly higher total at% of the COOH and OH groups of CN3-Pt and CN9-Pt was expected to reduce $Pt^{4+}$ to a lower valence state and increase effective charge transfer owing to its electron storage capacity [22]. Due to spin-orbit coupling, the Pt 4f spectra of photocatalysts could be split into $4f_{5/2}$ and $4f_{7/2}$. The peaks at 71.2 and 74.5 eV fit with the characteristics of metallic Pt, confirming the XRD data [42,43], and $Pt^{2+}$ and $Pt^{4+}$ appeared at higher binding energies of 72.7, 76.1, 74.9, and 78.2 eV, respectively [44,45]. The Pt 4f spectra of CN3-Pt and CN9-Pt only demonstrate the presence of $Pt^{2+}$ and $Pt^0$ species, revealing the complete $Pt^{4+}$ reduction in the hydrogen reduction process. Hence, the strong interaction between Pt and CN was clarified.

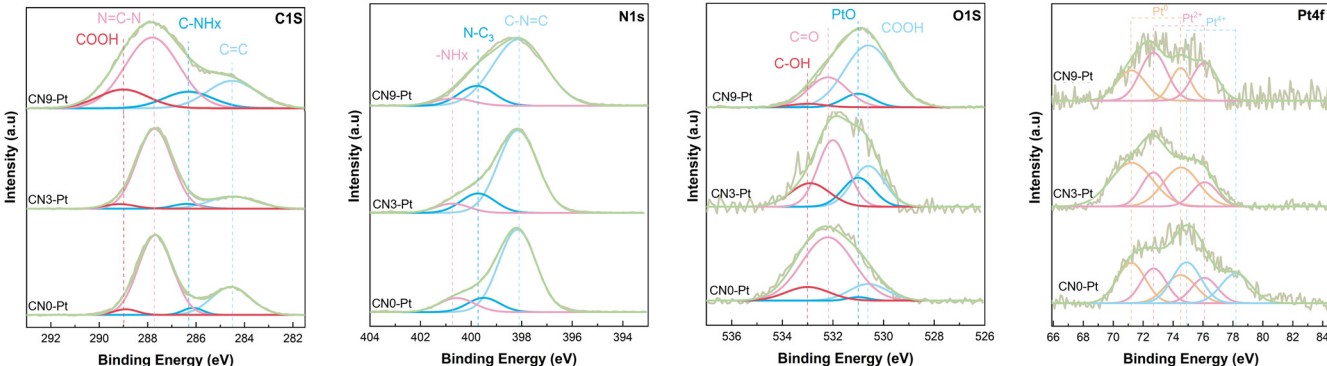

**Figure 2.** The XPS spectra of C 1s, N 1s, O 1s, and Pt 4f of CNx-Pt photocatalysts.

**Table 1.** Elemental analysis, Pt content (wt.%), and the atomic percentage of O-containing functional groups.

| Samples | wt.% | | | | | C/N | Atomic % | | |
|---|---|---|---|---|---|---|---|---|---|
| | N [1] | C [1] | H [1] | O [1] | Pt [2] | | C = O [3] | −OH [3] | −COOH [3] |
| **CN0-Pt** | 61.19 | 36.71 | 1.51 | 6.33 | 1.34 | 0.60 | 71.57 | 12.90 | 13.86 |
| **CN3-Pt** | 58.87 | 33.96 | 1.78 | 9.22 | 1.68 | 0.58 | 38.89 | 17.90 | 25.85 |
| **CN9-Pt** | 51.77 | 29.06 | 2.91 | 17.13 | 0.85 | 0.56 | 25.13 | 2.48 | 63.46 |

[1] Obtained from EA analysis. [2] Obtained from inductively coupled plasma–optical emission spectroscopy (ICP-OES) measurement. [3] Obtained from O1s XPS data.

The elemental analysis further confirmed the co-existence of Pt and CN, as displayed in Table 1. Besides that, the C/N ratio of CNx-Pt slightly decreased, suggesting that the chemical structure of CN was mostly unaltered. In the XPS data, the COOH group amount was determined by the conversion of C = O groups. Notwithstanding, the O content of CN9-Pt significantly increased to 17.13 wt.%, almost two times higher than that of CN3-Pt (9.22 wt.%) and CN0-Pt (6.33 wt.%), indicating that the O introduction may correspond to the extension of melem unit formation in the CN9-Pt structure. The O-containing groups were absorbed into the structural defects caused by the melem unit, increasing the O-containing functional groups in CN9-Pt during the hydrothermal process, and could enhance the photocatalytic activity [46].

### 2.2. Morphological Properties

The surface morphology of as-prepared CNx-Pt samples was investigated via FE-SEM, as displayed in Figure 3. As shown in Figure 3a, an overlapping lamellar structure was formed in CN0-Pt by stacking layers on top of each other, resulting in thicknesses of about 200–300 nm. The CN3-Pt surface had a more flake-like layer due to the regularly exfoliated lamellar structure under the hydrothermal treatment (Figure 3b). Figure 3c,d illustrate the results of the longer hydrothermal process time for CN9-Pt. The exfoliated g-$C_3N_4$ transformed into a hexagonal rod-like structure with a uniform length and width of about 1000 and 250 nm, respectively. In Figure S1a–e, the distribution of the C, N, O, and Pt components of CN9-Pt are analyzed and demonstrated by energy-dispersive spectroscopy elemental mapping. As a result, the melem unit extension in CN, changing the morphological structure of CN9-Pt into a rod-like structure, led to improved interactions between g-$C_3N_4$ and Pt, leading to an even Pt distribution (Figure S1e).

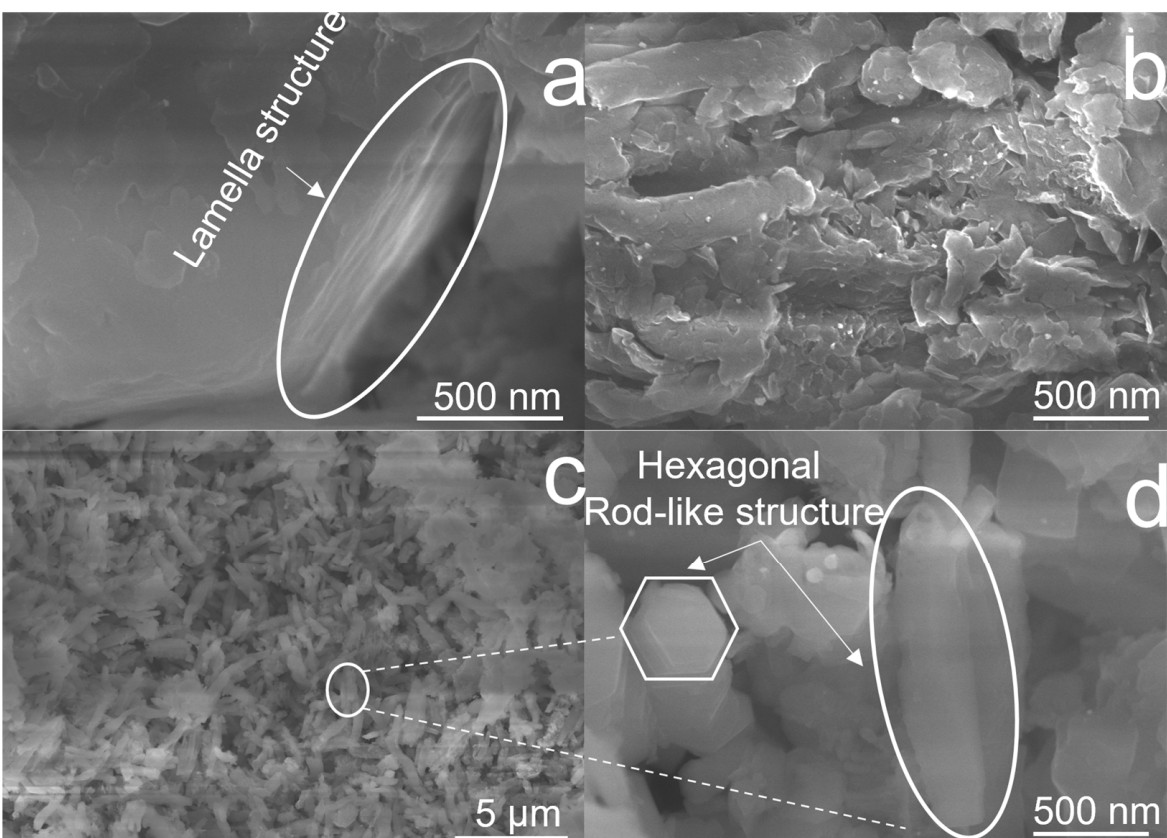

**Figure 3.** FE-SEM images of (**a**) CN0-Pt, (**b**) CN3-Pt, and (**c,d**) CN9-Pt.

### 2.3. Optical Properties

The ultraviolet–visible (UV–Vis) diffuse reflection spectra (DRS) of CNx-Pt were studied, as depicted in Figure 4A. CNx-Pt had a strong absorption peak at 325 nm in the ultraviolet region and a broad tail in the visible-light region. Compared to CN0-Pt and CN3-Pt, CN9-Pt showed a red-shift edge and a higher adsorption shoulder in the visible-light region (325–800 nm), suggesting a substantial light-harvesting capacity improvement and effective photocatalytic activity. The higher adsorption capacity of CN9-Pt could be presumably attributed to the transformation of the lamellar structure to the hexagonal rod-like structure of CN during the long-time hydrothermal treatment [47]. Further, the band gap estimated from the Tauc plot for CN0-Pt, CN3-Pt, and CN9-Pt significantly decreased by 2.82, 2.78, and 2.58 eV, respectively, as shown in Figure 4B.

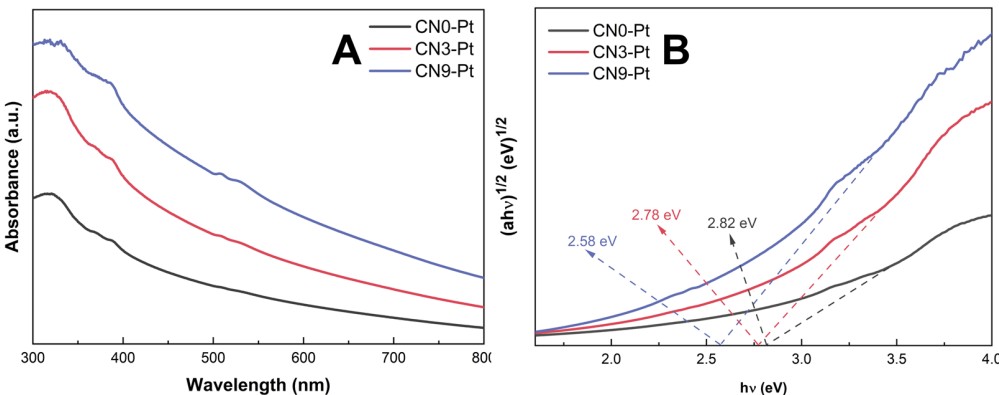

**Figure 4.** (**A**) UV–Vis DRS spectra and (**B**) Tauc plots for band-gap energy of photocatalysts.

Photoluminescence (PL) was investigated with an excitation wavelength of 350 nm to investigate the separation and recombination rate of the photogenerated electron–hole pairs of the as-prepared samples, and the corresponding results are displayed in Figure 5A. Herein, a prominent emission peak was observed at 462 nm for CNx-Pt. However, CN0-Pt manifested the highest peak intensity and broadband, indicating the fast recombination of charge carriers in the CN structure. The peak intensity of CN9-Pt was considerably reduced, suggesting the slow recombination rate of charge carriers after long-time hydrothermal treatment. Therefore, the rod-like structure formed by building a melem unit could inhibit the recombination of electron–hole pairs and thus enhance the photocatalytic activity [48,49].

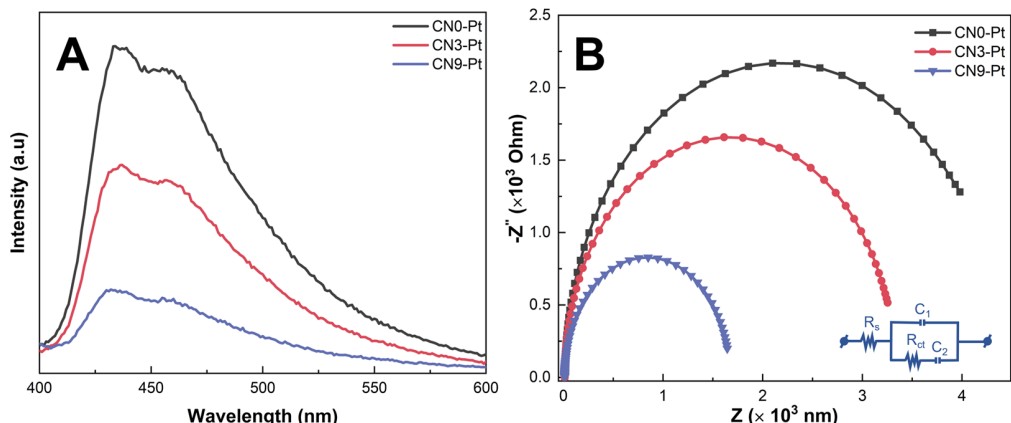

**Figure 5.** (**A**) PL spectra and (**B**) the EIS Nyquist plots of the CNx-Pt photocatalysts.

EIS was analyzed to further demonstrate the charge separation efficiency of photogenerated carriers. In general, a smaller arc radius in the EIS plot implies a higher charge-transfer efficiency. As illustrated in Figure 5B, the arc radius of the semicircle of the Nyquist diagram of CN9-Pt is smaller than those of CN0-Pt and CN3-Pt, reflecting the high conductivity and effective charge transfer. It is noteworthy that the arc radius of CNx-Pt gradually declined in the order of CN0-Pt > CN3-Pt > CN9-Pt, accompanied by an increase in O-containing functional groups. The parameters ($R_s$, $R_{ct}$, $C_1$, and $C_2$) fitted to the Nyquist plots are listed in Table S1 (see Supplementary Materials). This suggests that extending melem units in the rod-like structure after the long-time hydrothermal treatment could facilitate the introduction of O-groups, enhancing charge carrier transfer.

### 2.4. Hydrogen Evolution Performance

The hydrogen evolution performance of CNx-Pt was examined under visible-light illumination to study the role of morphological changes in the improved photocatalytic activity. As shown in Figure 6A, hydrogen production over CN3-Pt (151.4 mmol. $g_{Pt}^{-1}$) was higher than that over CN0-Pt (129.5 mmol. $g_{Pt}^{-1}$). In addition, the $H_2$ evolution over CN9-Pt was the highest, 220.2 mmol. $g_{Pt}^{-1}$, indicating the effect of the rod-like structure on the hydrogen evolution performance. Beyond that, the $H_2$ evolution rate of CNx-Pt increased in the order of CN0-Pt < CN3-Pt < CN9-Pt, as 25.91 < 30.23 < 43.91 (mmol. $g_{Pt}^{-1}.h^{-1}$), as seen in Figure 6B. Wide-range light absorption in the visible-light region, the slow recombination of photogenerated electron holes, and impressively high charge-transfer efficiency can explain the notably increased photocatalytic performance of CN9-Pt. The melem unit extension in CN9-Pt could change the morphology into a rod-like structure, improving the electron transfer. Moreover, the melem unit could obviously introduce more O-containing functional groups via edge defects. Furthermore, the photocatalytic stability of CN9-Pt was investigated in three-run test experiments, as illustrated in Figure S1 (see Supplementary Materials). After a three-run test, $H_2$ production was not decreased, demonstrating excellent photocatalytic stability and reusability for visible-light-driven $H_2$ generation.

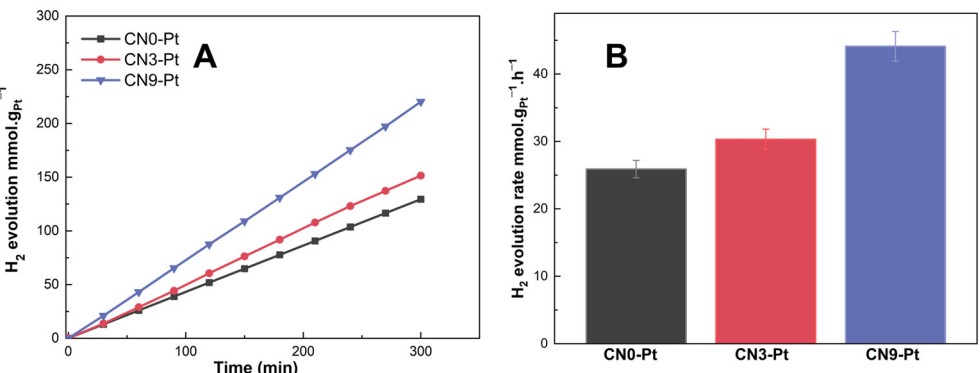

**Figure 6.** (**A**) Hydrogen evolution of CNx-Pt ($\mu$mol. $g_{Pt}^{-1}$) over time (min) and (**B**) hydrogen evolution rate of CNx-Pt ($\mu$mol.$g_{Pt}^{-1}. h^{-1}$).

The morphological changes in CN0-Pt and CN9-Pt and their relative physicochemical properties are perspicuously depicted in Figure 7 based on the above results. At first, the lamellar structure of the g-$C_3N_4$ material was formed by the layer-by-layer stacking of CN, hindering the electron transfer and revealing fewer O-containing functional groups for the $H_2$ evolution reaction. After a long-time hydrothermal treatment, the lamellar structure transformed into a rod-like structure with a hexagonal crosscut, accompanied by extended melem units and supplementary O-functional groups on the CN9-Pt structure. The hexagonal rod-like structure provided more active sites for photogenerated electrons, resulting in a significant reduction in electron recombination and charge-transfer efficiency, which was demonstrated in the PL and EIS data.

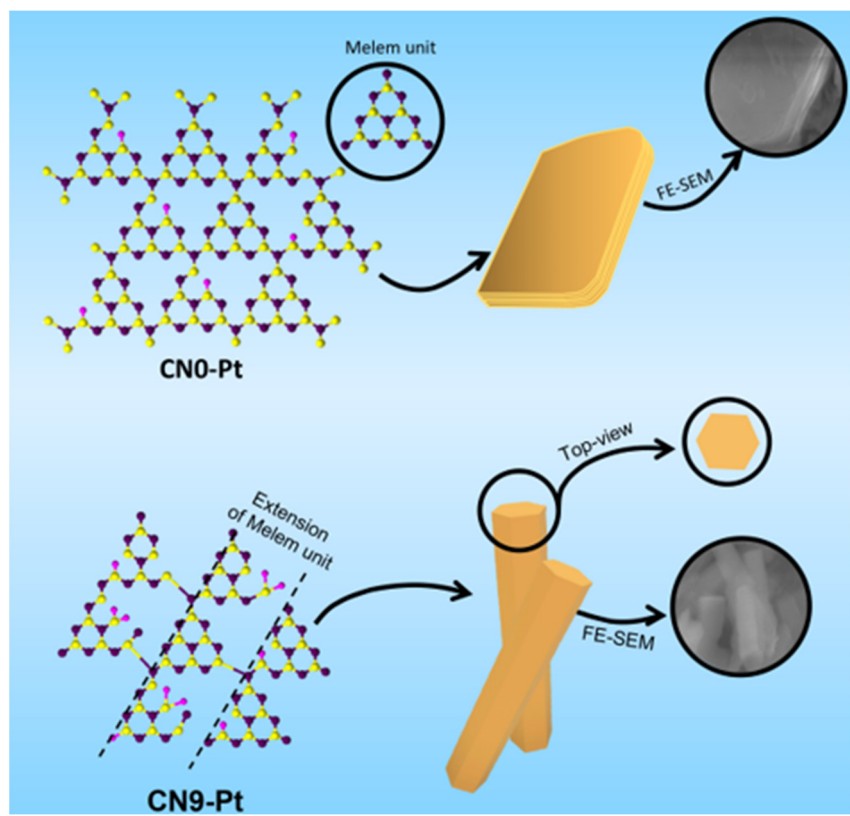

**Figure 7.** A schematic diagram for physicochemical and morphological modifications of CNx-Pt.

## 3. Materials and Methods

### 3.1. Synthesis of CNx and CNx-Pt

Thermal polycondensation was utilized for the synthesis of pure g-$C_3N_4$. Thiourea (30 g) ($CH_4N_2S$, ≥99%, supplied by Sigma-Aldrich Korea, Seoul, Republic of Korea), as the precursor, was loaded in an aluminum-foil-covered crucible and then heated in a muffle furnace to 550 °C for 4h at a heating rate of 5 °C/min under an air atmosphere. The as-prepared bulk g-$C_3N_4$ sample was designated as CN.

After being dissolved in 100 mL of deionized (DI) water, 1 gram of CN powder was sonicated for two hours at room temperature. The CNx samples (x = 0, 3, and 9 h, where x is the hydrothermal period) were placed in a 150 mL TeflonTM autoclave, which was heated to 180 °C (heating rate = 5 °C/min) in an oven. After being cooled to room temperature, the resulting materials were repeatedly rinsed with DI water.

A magnetic stirrer was used to dissolve the CNx samples in 1000 mL of DI water for 30 min. The $H_2PtCl_6.6H_2O$ amount added to the mixture was calculated to set the initial content of Pt at 2 wt.%. Pt on CNx was processed with hydrogen reduction for 60 min under a 30 rpm $H_2$ flow rate at 60 °C. The precipitates were collected, repeatedly washed with DI water, and freeze-dried for additional characterization in the final step. The materials were CN0-Pt, CN3-Pt, and CN9-Pt, according to their CNx precursors.

### 3.2. Characterization

The Pt element composition of CNx-Pt samples was validated using an inductively coupled plasma–optical emission spectrometer (ICP-OES) (700-ES; Varian Australia Pty. Ltd., Mulgrave, Australia). The morphologies of the CNx-Pt samples were investigated using field-emission scanning electron microscopy (FE-SEM) (Quanta 200 FEG, FEI Corp., Hillsboro, OR, USA). A Cu-K$\alpha$ X-ray source with a wavelength of $\lambda$ = 1.5415 Å (D/MAZX 2500 V/PC high-power diffractometer; Rigaku, Tokyo, Japan) was used to record the XRD patterns of the samples at a scan rate of 2° (2θ)/min. A Fourier transform infrared (FTIR)

transmittance spectrometer (Nicolet™ 380 spectrometer, Nicolet™ iS5 with an iD1 transmission accessory; Thermo Scientific™, Waltham, MA, USA) was employed to examine the functional groups of the prepared photocatalysts. Thermo Scientific's K-Alpha system (Waltham, MA, USA) was utilized for XPS. A UV–Vis diffuse reflectance spectrometer (SPECORD®210 Plus spectroscope; Analytik Jena, Jena, Germany) and photoluminescence (PL) measurements (Cary Eclipse fluorescence spectrophotometer; Agilent, Santa Clara, CA, USA) were used to analyze the optical properties of the photocatalysts. In PL measurements, a 473 nm diode laser was used to monitor the PL spectra at ambient temperature.

### 3.3. Photocatalytic Hydrogen Evolution Test

For the photocatalytic $H_2$ evolution experiments, a solar simulation system with a light source consisting of an LED lamp with a light intensity set to 100 mW.cm$^{-2}$ was used. Each photocatalyst (30 mg) was then dissolved for 30 min with high-purity Argon purging in the dark in 100 mL of DI water. As a hole scavenger, 10 mL of triethanolamine (TEOA) was incorporated into the solution. Argon was used to purge the reactor system before starting $H_2$ evolution tests. An online gas chromatograph with an installed thermal conductivity detector was utilized to examine the gas products. The same setup in 5 h cycles was used for the photocatalyst stability tests. At the beginning of each cycle, an extra amount of TEOA (10 mL) was added to the solution, and Argon was again used to purge the reactor system.

### 3.4. Photoelectrochemical Test

In addition, an impedance analyzer (VSP series; Bio-Logic Science Instruments, Seyssinet–Pariset, France) was utilized to measure the EIS data. After a 10 min interval under the illumination of a 3 W visible-light bulb, a frequency range of 0.01 Hz to 100 kHz with an amplitude of 10 mV and a direct current potential of +0.8 VSCE were used in the test. Then, 20 mg of the catalyst powder ground in a mortar was mixed with 2 mg of active carbon for 20 min to produce a fine powder mixture. The powder was then incorporated into 100 μL of isopropanol 99.7% and 30 μL of Nafion 5 wt.% (both from Sigma-Aldrich Korea, Gyeonggi, Korea). Next, 10 mL of potassium hydroxide 1 M (KOH) was used as the electrolyte in a 3-electrode setup. The reference and counter electrodes were a RE-1BP (Ag/AgCl) electrode and a platinum wire. A 6 mm standard-type glassy carbon electrode with 10 μL of the sample (1 μL of suspension per time by a micro-pipette) loaded on it was used as the working electrode.

### 4. Conclusions

In conclusion, the facile hydrothermal modification of g-$C_3N_4$-Pt was examined in this study, and the impact of the modification on $H_2$ production activity was evaluated. As a result of a long-time hydrothermal process, the layer-by-layer structure of CN was clearly altered by its transformation to a rod-like structure via the extension of melem units bearing more oxygen-containing functional groups. It was demonstrated that the morphological modification of as-prepared CN9-Pt samples exhibited enhanced visible-light absorption, faster charge carrier separation, and the inhibition of the recombination rate of electron–hole pairs. In addition, a greater number of O-functional groups were introduced into CN9-Pt, which facilitated the $Pt^{4+}$ reduction to metallic Pt and $Pt^{2+}$ for the $H_2$ evolution reaction. Consequently, the morphological modification of CN9-Pt manifested the highest $H_2$ evolution performance and stability over the recycle times.

**Supplementary Materials:** The following supporting information can be downloaded at: https://www.mdpi.com/article/10.3390/catal13010092/s1, Figure S1: Elemental mapping of (b) C, (c) N, (d) O, and (e) Pt of (a) CN9-Pt; Figure S2: Photocatalytic stability test of $H_2$ production for CN9-Pt; Table S1: Fitted parameters of CN0-Pt, CN3-Pt, and CN9-Pt for the Nyquist plots.

**Author Contributions:** T.V.A.H., investigation, writing—original draft, and editing; P.A.N., analysis and data curation; E.W.S. supervised the work and refined this manuscript. All authors have read and agreed to the published version of the manuscript.

**Funding:** This research was funded by the National Research Foundation of Korea (NRF), funded by the Korean government (MIST; 2020R1A4A4079954 and 2021R1A2B5B01001448).

**Data Availability Statement:** The data presented in this study are available on request from the corresponding author.

**Conflicts of Interest:** The authors declare no conflict of interest.

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
