# Peer review of "Effect of Morphological Modification over g-C3N4 on Photocatalytic Hydrogen Evolution Performance of g-C3N4-Pt Photocatalysts"

_catalysts, doi:10.3390/catal13010092_

Round 1

Reviewer 1 Report

so many errors are observed starting  from abstract presentation.

Authors have to take care such mistakes like 1) modi-fied, 2)inves-tigate, 3) spectro-scopy...etc.

The entire manuscript should be well presented in terms of good English

Reviewer 2 Report

The authors employ a simple hydrothermal treatment to prepare tunable CN-Pt catalysts in the manuscript. A series of characterization methods confirmed the successful preparation of the photocatalysts, and the CN-Pt catalyst has excellent photocatalytic hydrogen production activity. The catalyst design is novel, and the article is logical and innovative.

1. In Fig. 1a, the authors explain that the expansion of the melem group in the CN structure causes the characteristic peak of CN at 13° to shift to 10.6°. Why? Authors are required to give necessary references.

2. In the C1s spectrum, the areas of the two peaks belonging to COOH and C-NHx in the CN9-Pt sample increased significantly. Why?

3. The markers in the XPS spectrum are challenging to see, and the author needs to change the appropriate colour.

4. The author needs to supplement the photocatalytic H2 production performance of pure CN, which illustrates the advantages of a Pt-supported CN catalyst.

5. The energy band position is crucial to the possibility of photocatalytic H2 production. The authors should provide the necessary energy band position and give a detailed photocatalytic mechanism.

6. Suggestion: consider citing some similar photocatalytic research reports. Such as Liu C, Mao S, Wang H, et al. Chemical Engineering Journal, 2022, 430: 132806; Liu C, Mao S, Shi M, et al. Journal of Hazardous Materials, 2021, 420: 126613; Chemical Engineering Journal, 2022, 449: 137757.

Reviewer 3 Report

The article describes the improvement of the photocatalytic properties of graphite-like carbon nitride by conventional hydrothermal treatment. The idea is not new and quite simple, but, in principle, good results have been obtained. I have only a number of minor remarks.

1. Fig. 3. Element mapping is of a very bad quality and gives no information.

2. Value 1514.48 umol g-1 h-1 has an excess of significant digits, 1510 would be more better.

3. Unit umol g-1 h-1 per wt% of Pt is very strange. It would be better to calculate it per gram of catalyst or per gram of Pt.

4. Fig. 6. The error of measurement should be shown.

5. The main note is that the improvement of catalytic properties by hydrothermal treatment is rather sparingly written and should be expanded upon. Moreover, the principle of action of the deposited platinum must be described.

6. The activities obtained in this work should be compare with recently published data on Pt/g-C3N4 photocatalysts.

Round 2

Reviewer 2 Report

Authors addressed very well most of my comments. Paper could be published now.